# Association of pregnancy complications/risk factors with the development of future long-term health conditions in women: overarching protocol for umbrella reviews

Megha Singh [1], Francesca Crowe [1], Shakila Thangaratinam,[2,3] Kathryn Mary Abel,[4,5] Mairead Black,[6] Kelvin Okoth,[1] Richard Riley,[7] Kelly-Ann Eastwood,[8,9] Holly Hope,[5] Steven Wambua,[1] Jemma Healey,[6] Siang Ing Lee [1], Katherine Phillips [1], Zoe Vowles,[10] Neil Cockburn,[1] Ngawai Moss,[11] Krishnarajah Nirantharakumar [1]

For numbered affiliations see end of article.

**Correspondence to**
Dr Megha Singh;
m.singh.6@bham.ac.uk

## ABSTRACT

**Introduction** With good medical care, most pregnancy complications like pre-eclampsia, gestational diabetes, etc resolve after childbirth. However, pregnancy complications are known to be associated with an increased risk of new long-term health conditions for women later in life, such as cardiovascular disease. These umbrella reviews aim to summarise systematic reviews evaluating the association between pregnancy complications and five groups of long-term health conditions: autoimmune conditions, cancers, functional disorders, mental health conditions and metabolic health conditions (diabetes and hypertension).

**Methods and analysis** We will conduct searches in Medline, Embase and the Cochrane database of systematic reviews without any language restrictions. We will include systematic reviews with or without meta-analyses that studied the association between pregnancy complications and the future risk of the five groups of long-term health conditions in women. Pregnancy complications were identified from existing core outcome sets for pregnancy and after consultation with experts. Two reviewers will independently screen the articles. Data will be synthesised with both narrative and quantitative methods. Where a meta-analysis has been carried out, we will report the combined effect size from individual studies. For binary data, pooled ORs with 95% CIs will be presented. For continuous data, we will use the mean difference with 95% CIs. The findings will be presented in forest plots to assess heterogeneity. The methodological quality of the studies will be evaluated with the AMSTAR 2 tool or the Cochrane risk of bias tool. The corrected covered area method will be used to assess the impact of overlap in reviews. The findings will be used to inform the design of prediction models, which will predict the risk of women developing these five group of health conditions following a pregnancy complication.

**Ethics and dissemination** No ethical approvals required. Findings will be disseminated through publications in peer-reviewed journals and conference presentations.

## STRENGTHS AND LIMITATIONS OF THIS STUDY

⇒ These umbrella reviews will combine the results from existing systematic reviews in the topic and compile in a single document.
⇒ These reviews will include reviews from all languages without any restriction of language.
⇒ Screening of studies and quality assessment of the reviews will be independently done by both reviewers. Overlap of the reviews will be taken into account.
⇒ The same data extraction and quality appraisal forms will be used across the reviews and will be piloted before use.
⇒ A limitation is that there might not be an existing systematic review for a few of the rare conditions and so these will not be included in the umbrella review.

## INTRODUCTION

Although maternal deaths have decreased globally by 38% in the last two decades due to improved and more accessible medical care, the occurrence of pregnancy complications has seen an increasing trend. For instance, worldwide incidence of hypertensive disorders of pregnancy increased by 11% over the past 20 years going from 16 to 18 million. The prevalence of gestational diabetes has also increased during the same time period alongside rising levels of obesity and inactivity.[1–5] Prior analyses have consistently identified environmental risk factors such as air pollution, poverty, alcohol intake, diet, smoking, obesity and weight gain in pregnancy with gestational diabetes and pre-eclampsia in the mother and preterm birth and low birth weight among the offspring. Both common

mental health disorders (ie, depression and anxiety) and serious mental health disorders (affective and non-affective psychotic disorders) are associated with poor health in pregnancy. Polycystic ovarian syndrome increases the risk of gestational diabetes and periodontal disease has also been identified as a potential risk factor for preterm birth and low birth weight.[6–12]

Globally, there were 295 000 maternal deaths in 2017 that were attributed to preventable causes related to pregnancy and childbirth, equivalent to 810 deaths per day.[13 14] Complications arising during pregnancy and childbirth continue to be leading causes of maternal deaths, the most common of which are postpartum haemorrhage or hypertensive disorders during pregnancy.[15] Most pregnancy complications like hypertensive disorders and gestational diabetes resolve after birth; however, they are associated with an increased risk of complications in future pregnancies as well as long-term physical and mental health conditions. During pregnancy, the maternal organs undergo significant physiological changes, such as increased cardiac output and inflammatory response due to the complications or risk factors, which can be reactivated by age-related changes in later life, resulting in development of long-term health conditions.[16–19] Women who have experienced pre-eclampsia have an increased risk of developing type 2 diabetes along with increased risk of hypertension, ischaemic heart disease, stroke and venous thromboembolism.[20–23] Depression and anxiety disorders are common after miscarriage, stillbirth and preterm births.[24–26] Studies have also identified pregnancy complications such as miscarriage and preterm birth are associated with future risk of breast cancer.[27 28]

While it is well established that gestational diabetes increases the risk of developing type 2 diabetes by tenfold, it is also associated with an increased risk of cardiovascular disease and cancer in later life.[29 30] Even complications often considered less serious such as hyperemesis gravidarum were associated with a 70% increased risk of developing rheumatoid arthritis. A relationship between miscarriage and future development of autoimmune diseases have also been identified.[31 32] There are systematic reviews, which have shown the relationship between pregnancy complications and risk factors and future development of health conditions, but there is a need for further research to generate robust evidence which can further be utilised for the betterment of maternal health postpregnancy.

We have previously published an umbrella review studying the association between pregnancy complications and cardiovascular diseases.[33] In this umbrella review protocol, we describe our objectives and methods to investigate the associations of pregnancy complications and five other groups of long-term health conditions, namely autoimmune conditions, cancers, functional conditions, mental health conditions and metabolic conditions (diabetes and hypertension). The umbrella reviews will aid in collating the necessary evidence to delineate the association between adverse pregnancy complications and subsequent occurrence of these health conditions. This will help develop early detection strategies and inform the prognostic factors that will be used to develop risk prediction models to predict future occurrence of these long-term health conditions.

## Protocol development

This umbrella review protocol was developed following Joanna Briggs Institute methodology for umbrella reviews.[34] The reporting was done using the Preferred Reporting Items for Systematic Reviews and Meta-Analyses Protocols (PRISMA-P) guidelines (online supplemental table 1).[35] The PRISMA-P has been registered with Prospero (registration number CRD42022323718). In line with the Prospero registration the start date was April 2022. Any deviations from the protocol will be reported in detail in the umbrella reviews.

## Aims and objectives
### Aims

These umbrella reviews will identify, appraise and consolidate higher level evidence in the form of systematic reviews with or without meta-analyses into a single readable/usable document and provide recommendations for future research.

### Objectives

To identify and appraise higher level evidence (systematic reviews and meta-analyses) reporting the associations between pregnancy complications/risk factors (box 1) and future risk of
1. Autoimmune conditions.
2. Cancers.
3. Functional health conditions.
4. Mental health conditions.
5. Metabolic health conditions.

If there are no systematic reviews for specific exposures and outcomes, we may consider conducting a separate systematic review that would be subject to peer review. We will not include primary studies in our umbrella reviews. The health conditions included in each of the above groups are listed below (table 1).

## RESEARCH PLAN/METHODS

Each umbrella review aims to identify, appraise, combine and synthesise all the available evidence for each outcome. The umbrella reviews will primarily include systematic reviews with or without meta-analyses.[36 37] Where there is no existing systematic review, the research team will undertake a scoping review to assess whether to conduct a new separate systematic review. The reporting of the reviews will be done using the PRISMA guidelines.[38]

## Study design

Systematic reviews of observational and interventional studies with or without meta-analysis that have assessed the association between the pregnancy complications/risk factors and future risk of long-term health conditions

## Box 1 Pregnancy complications and risk factors

1. Pregnancy loss
   – Miscarriage/recurrent miscarriage/spontaneous pregnancy loss.
   – Stillbirth.
2. Hypertensive disorders of pregnancy
   – Gestational hypertension.
   – Pre-eclampsia- early or late onset.
   – Recurrent pre-eclampsia.
   – Eclampsia.
   – Haemolysis, elevated liver enzymes and low platelets syndrome.
3. Placental disorders
   – Placenta previa.
   – Placental abruption.
   – Placenta accreta.
   – Placenta percreta.
4. Hyperemesis gravidarum.
5. Gestational diabetes mellitus.
6. Ectopic pregnancy.
7. Molar pregnancy/choriocarcinoma.
8. Multiple pregnancy/twin-pregnancies/multiple gestation.
9. Obstetric haemorrhage (post partum).
10. Pre-term birth/recurrent preterm birth.
11. Mode of birth: caesarean, instrumental.
12. Low birth weight
    – Low birth weight.
    – Small for gestational age.
    – Intrauterine growth retardation/intra-uterine growth restriction.
    – Fetal growth restriction.
13. Postpartum depression.
14. Puerperal psychosis.
15. Perineal trauma—third-degree and fourth-degree tear.
16. Obstetric cholestasis.
17. Pelvic girdle pain.

in women will be considered for these umbrella reviews. As the purpose of an umbrella review is to identify and synthesis evidence from systematic reviews, we will include all relevant systematic reviews.[39 40] Some of these systematic reviews may include interventional studies. Any secondary analysis of data collected in intervention studies around exposure and outcome of interest will be included although the exposure does not have to be the intervention in the intervention study.

## Population
The population will include pregnant women, no age restriction.

## Exposures
The exposure of interest are adverse pregnancy complications/risk factors which will be identified by scoping searches and consultation with experts. The list of pregnancy complications and risk factors identified are listed below in box 1 and the definitions are listed in online supplemental table 2. Depending on the outcome in question, a few of the exposures might not be considered.

## Comparator
Pregnant women without the exposure of interest will be the comparator group.

## Outcomes
The outcomes of interest are the following five groups of long-term health conditions described in table 1.

## Search strategy
We will conduct searches in Medline, Embase and the Cochrane database of systematic reviews from inception. We will be looking for systematic reviews with or without meta-analysis that examine the associations between pregnancy complications and future risk of the long-term health condition in women. There will be no restriction of language or setting when selecting the studies.

The search strategy will be developed using subject headings and free text keywords using the concepts for pregnancy complication/risk factors (listed in online supplemental table 3) and for the specific disease in the group of long-term health conditions to be studied (listed in online supplemental table 4). The reference list of included studies will also be searched. The search terms and detailed search strategy for Medline are provided in online supplemental table 3, 4 and 5.1–5.5, respectively. These will be modified for use in other databases.

| Table 1 | List of health conditions (outcomes) |
| --- | --- |
| **Conditions** | |
| Autoimmune conditions | Psoriasis, vitiligo, alopecia areata, systemic lupus erythematosus, psoriatic arthritis, ankylosing spondylitis, rheumatoid arthritis, Sjögren's syndrome, systemic sclerosis, coeliac disease, Crohn's disease, ulcerative colitis, (inflammatory bowel disease), multiple sclerosis, grave's disease, Hashimoto's disease, type 1 diabetes mellitus, myasthenia gravis, Addison's disease |
| Cancers | Lung, breast, colorectal, cervix, cancer of unknown primary, pancreas, ovary, uterus/endometrium, brain, (other central nervous system) and intracranial tumours, liver, melanoma skin cancer, lymphoma, kidney, thyroid, leukaemia. |
| Functional conditions | Fibromyalgia, chronic pain, chronic back pain, tension headache, irritable bowel syndrome, interstitial cystitis, vulvodynia, irritable bowel syndrome |
| Mental health | Serious mental illness-affective psychosis (bipolar, mania) and non-affective psychoses (schizophrenia, paranoia and psychoses nos.) Common mental illness-mood disorders (depression, dysthymia) and neurotic (generalised anxiety, panic, post-traumatic stress disorder, obsessive compulsive disorder etc) disorders |
| Metabolic conditions | Type 2 diabetes mellitus and hypertension. |

A two-pronged approach will be adapted, first search strategy will be restricted to systematic reviews to identify which pregnancy complications/risk factors will have evidence from systematic reviews. Second, we will run a search strategy for primary studies if an update of the review is required. Searches will be updated periodically to identify newly published systematic reviews.

### Study selection

Once the literature search is completed, reference management software will be used to manage the identified literature (eg, Endnote, Mendeley or Reference Manager). After removing duplicate studies, two reviewers will independently conduct the title and abstract screening and ineligible studies will be excluded. Full-text screening of eligible studies will be conducted by two reviewers independently and a third senior reviewer will be consulted to resolve any discrepancy. The full text will be translated if non-English language studies are identified. For non-English language articles, the authors will be consulted if they have expertise in the language. Followed by contacting university language departments and fellow researchers. If no one is identified with expertise in the language, then a professional will be hired for the translation.[41] The list of excluded studies will be maintained with the reasons for exclusion documented. The details of the steps involved in study selection will be reported using the PRISMA flow chart.

### Exclusion criteria

The following types of publications will not be included: protocols, review articles, conference abstracts, guidelines, consensus, documents or expert position papers, summaries, comments, letters and brief reports. The reviews that include theoretical studies or text or opinion as their primary source of evidence will not be included.[34 42]

### Quality assessment

Two reviewers will perform the quality assessment of the reviews using AMSTAR 2 tool independently. Out of the 16 points of the AMSTAR 2 tool, 1 point will be awarded for each of the criteria met.[43] The reviews will be rated as high, moderate, low or critically low quality. The critical domains will include protocol registration, literature search detailed and including grey literature, risk of bias assessment of included primary studies, meta-analysis conducted appropriately, risk of bias considered in reporting/interpretation of results and reporting of publication bias will be taken into consideration for rating the review. The reviews will not be excluded even if they are rated as being low quality. Reviews that do not mention the quality assessment of primary studies might be excluded but this will be at the reviewer's discretion. To resolve any disagreements, a third reviewer will be consulted.

### Update of existing reviews

Up to 50% of systematic reviews are out of date after 5.5 years.[44] The recommended methods for updating existing systematic reviews and meta-analyses will be used where a need for update is identified and the update of the review will only be considered by authors based on these guidelines.[45–48] In case the reviews need to be updated, only high and moderate quality systematic reviews will be eligible for updates.[45]

### Overlap

Overlapping reviews refers to a situation where two or more reviews examine the same research question and may include the same primary studies. The degree of overlap will be presented graphically using a citation matrix. A citation matrix is a plot of the included primary studies in the rows and systematic reviews in the columns. Overlap will be quantified by the method of corrected covered area.[49 50] Quality rating will be used as the selection criteria where higher-quality reviews will be considered over lower-quality reviews.

### Data extraction

A standardised data extraction form will be used. The data extraction form will be piloted prior to use. Data will be extracted from the final list of included studies which will then be checked by the second reviewer. Two reviewers will be involved in data extraction. After the first reviewer has completed the data extraction, the second reviewer will check the data extraction sheet and provide any comments. Any differences will be resolved by discussion and a third reviewer will be consulted if necessary.

The data will be extracted using the following template using Microsoft Excel.
1. Study ID.
2. Author/s.
3. Year of publication.
4. Geographical area.
5. Aim of the review.
6. Database searched
7. Search period.
8. Population.
9. Heathcare setting.
10. Exposures.
11. Comparator.
12. Outcomes
13. Covariates
14. Study design(s).
15. Definition of exposure.
16. Definition of outcome.
17. Data synthesis method.
18. Quality assessment tool.
19. Quality of the included primary studies as assessed by review authors.
20. Number of studies included in qualitative analysis (narrative synthesis where meta-analysis was not done/possible by the review authors).
21. Number of meta-analyses.
22. Number of studies included in each meta-analysis.
23. Summary estimates of each meta-analysis and its related 95% CIs.
24. Author's conclusion.

25. Review limitation.
26. Additional comments.

Recently published studies will be added into the narration of the reviews where no meta-analysis has been performed.

## Data analysis

Study characteristics will be presented in a tabular format. The study will synthesise the data using both narrative and quantitative methods. Where a meta-analysis has been carried out, we will report the summary result. Where the review does not provide a summary result, we will explore the key findings and use these to inform a narrative overview of the key findings. Based on the data extracted from individual studies, systematic review or meta-analysis, data may need to be converted before pooling. For binary data, pooled ORs with 95% CIs will be presented. For continuous data, we will use the mean difference or standardised mean difference with 95% CIs. While analysing the results reported in the systematic reviews, there may be differences in population characteristics, so it might not be possible to combine all the results from the included studies. If we do combine them, we may present effect sizes according to certain population characteristics such as country/region to account for differences in nutritional status, etc. We will try to consider all heterogeneity including clinical, methodological and statistical and will ensure this is covered in the discussion section of the manuscripts of the umbrella reviews. The findings will be presented in forest plots to assess the heterogeneity of the study findings. The $I^2$ statistic will be used to evaluate statistical heterogeneity. All statistical analyses will be conducted using Stata, Microsoft Excel or R package. Age and ethnicity subgroup analysis will be considered where appropriate. Publication bias will be assessed both quantitatively and using a funnel plot where appropriate.[51] Grading of Recommendations, Assessment, Development and Evaluation tool will be used to estimate the strength of evidence.[52]

## Patient public involvement

Patient and public involvement representatives (KP and NM) were involved in formulating the research question and study design. They have also played a key role in collaboration with clinicians and researchers to identify and consider the list of pregnancy complications and health outcomes in the study. NM has been part of our regular meetings and also provided input to improve the manuscript. They will also play a key role in disseminating the results once the reviews have been undertaken.

## ETHICS AND DISSEMINATION

No ethical approvals required. Findings will be disseminated through publications in peer-reviewed journals and conferences.

### Author affiliations

[1]Institute of Applied Health Research, University of Birmingham, Birmingham, UK
[2]WHO Collaborating Centre for Global Women's Health, Institute of Metabolism and Systems Research, University of Birmingham, Birmingham, UK
[3]Department of Obstetrics and Gynaecology, Birmingham Women's and Children's NHS Foundation Trust, Birmingham, UK
[4]Medical and Human Sciences, Institute of Brain Behaviour and Mental Health, Manchester, UK
[5]Centre for Women's Mental Health, Faculty of Biology Medicine & Health, The University of Manchester, Manchester, UK
[6]Aberdeen Centre for Women's Health Research, School of Medicine, Medical Science and Nutrition, University of Aberdeen, Aberdeen, UK
[7]Centre for Prognosis Research, School of Primary, Community and Social Care, Keele University, Staffordshire, UK
[8]St Michael's Hospital, University Hospitals Bristol NHS Foundation Trust, Bristol, UK
[9]Centre for Public Health, Queen's University of Belfast, Belfast, UK
[10]Guy's and St. Thomas' NHS Foundation Trust, London, UK
[11]Patient and public representative, London, UK

**Acknowledgements** Patient representatives and MuM-PreDiCT team.

**Contributors** MS was responsible for drafting the initial manuscript. KN, FC, ST, MB, K-AE, HH, JH, SW, ZV, KP, NM, NC, SIL, ST, KO, KMA and RR were responsible for revising the manuscript critically for important intellectual content. The authors have approved the final submitted version and are accountable for all aspects of the work.

**Funding** This work was funded by the Strategic Priority Fund 'Tackling multimorbidity at scale' programme (grant number-MR/W014432/1) delivered by the Medical Research Council and the National Institute for Health and Care Research in partnership with the Economic and Social Research Council and in collaboration with the Engineering and Physical Sciences Research Council.

**Disclaimer** The funders have no role in development of this protocol.

**Competing interests** None declared.

**Patient and public involvement** Patients and/or the public were involved in the design, or conduct, or reporting, or dissemination plans of this research. Refer to the Methods section for further details.

**Patient consent for publication** Not applicable.

**Provenance and peer review** Not commissioned; externally peer reviewed.

**Data availability statement** Not applicaple.

### ORCID iDs

Megha Singh http://orcid.org/0000-0003-3680-7124
Francesca Crowe http://orcid.org/0000-0003-4026-1726
Kelvin Okoth http://orcid.org/0000-0002-2745-4083
Richard Riley http://orcid.org/0000-0001-8699-0735
Siang Ing Lee http://orcid.org/0000-0002-2332-5452
Katherine Phillips http://orcid.org/0000-0003-0674-605X
Krishnarajah Nirantharakumar http://orcid.org/0000-0002-6816-1279

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
