## [Reviewer comments · BMJ Open]

ARTICLE DETAILS

TITLE (PROVISIONAL)	The association of pregnancy complications/risk factors with the development of future long-term health conditions in women: overarching protocol for umbrella reviews
AUTHORS	Singh, Megha; Crowe, Francesca; Thangaratinam, Shakila; Abel, Kathryn; Black, Mairead; Okoth, Kelvin; Riley, Richard; Eastwood, Kelly-Ann; Hope, Holly; Wambua, Steven; Healey, Jemma; Lee, Siang Ing; Phillips, Katherine; Vowles, Zoe; Cockburn, Neil; Moss, Ngawai; Nirantharakumar, Krishnarajah

VERSION 1 – REVIEW

REVIEWER	Sanefuji, Masafumi Kyushu University, Pediatrics
REVIEW RETURNED	14-Sep-2022

GENERAL COMMENTS	The association of pregnancy complications/risk factors with the development of future health conditions in women: Overarching protocol for umbrella reviews This umbrella review that would start in March 2022 aims to summarise systematic reviews evaluating the association between pregnancy complications and five long-term health conditions of the women. This topic is very important for public health. However, there are several issues that should be addressed very carefully. There are lots of grammatical errors throughout the manuscript (e.g. space, period, upper or lower case letters). It is a bit difficult to follow as the orders of pregnancy complications and health conditions are inconsistent among the text and tables. In 'Objectives' (p5), 1) Metabolic health conditions, 2) Functional health conditions, 3) Autoimmune conditions, 4) Cancers, and 5) Mental health conditions are listed. However, Table 2 lists 'Mental health', 'Functional conditions',... , and Appendix Table 3 lists 'Autoimmune conditions', 'Metabolic health conditions'..., in order. Please keep consistency in order and terminology (e.g. 'Autoimmune conditions' (text) vs. 'Autoimmune disorders' (Table 2), 'Autoimmune health conditions' (Appendix Table 3)). 'Placental abruption' in Table 1 is not defined in Appendix Table 1. 'Irritable bowel syndrome' is categorised as 'Functional conditions' in Table 2 although it is listed as 'Autoimmune health conditions' in Appendix Table 3. The order of 'Pre-eclampsia-early and late onset' to 'Gestational hypertension' should be rearranged according to that of
---

	'Hypertensive disorders of pregnancy' in Table 1. [Appendix Table 1] 'PPH' should be spelled out in 'Obstetric/postpartum haemorrhage'. I'm wondering why analysis is limited to 'singleton baby' in 'Mode of birth'. Is multiple birth more likely to be born with Caesarean section? Spaces are needed between each item if applicable. For example, 'Fetal growth restriction' should be positioned to the same level to its explanation. Is 'Or' required at the first in 'Puerperal psychosis'? 'Pelvic girdle pain (PGP)' could be exchanged in order for 'Obstetric cholestasis' according to Table 1. [Appendix Table 2] The items 'Placenta previa' and 'Placenta accrete and placenta percreta' do not correspond to their explanations. 'Fetal growth retardation' listed in Table 1 could be added within the same square of 'Intra-uterine growth retardation/intra-uterine growth restriction'. '17. Pelvic girdle pain' in Table 1 is not seen. [Appendix Table 3] The order of disease listed should be rearranged exactly according to Table 2. For example, 'Systemic sclerosis' should be positioned just below 'Sjogren's syndrome', (not 'Sjogern's'). 'Addison's disease (in Autoimmune conditions)' and 'Oesophagus (in Cancers)' is not seen in Table 2. In reverse, 'colorectal' in Table 2 is not described in this appendix. In 'Mental health Conditions', 'non-affective psychosis' and 'neurotic disorders' in Table 2 are not defined, too.
--	--

REVIEWER	Schomburg, Lutz
REVIEW RETURNED	Charité Universitätsmedizin Berlin, Institute for Exper. Endocrinology 29-Sep-2022

GENERAL COMMENTS	The manuscript, titled "The association of pregnancy complications/risk factors with the development of future health conditions in women: Overarching protocol for umbrella reviews," describes in detail how the authors will compile and present the current evidence and data on the topic highlighted in the title. The protocol is of the highest scientific quality, the methods are clearly described, and the topic is of paramount importance. The reviewer found no deficiencies or inaccuracies in the text. However, a discussion of possible limitations would be desirable. How do the authors deal with different populations, genetic backgrounds, women with specific dietary preferences, and combinations thereof? For example, the trace element selenium has been shown to be important in pregnancy outcomes, child development, and postpartum thyroiditis and depression. However, this effect may be limited to populations with low baseline intakes of selenium, such as in many parts of Asia, Europe, and Africa, but notably not in the U.S., where most high-quality studies are conducted. Often such specific interactions are not taken into account in regular reviews or meta-analyses, yet they may become apparent in an umbrella review. A few words about how the authors will handle such oddities would certainly strengthen both the protocol and the final study.
--

REVIEWER	Meena , Jyoti All India Institute of Medical Sciences
REVIEW RETURNED	02-Oct-2022

GENERAL COMMENTS	This study will focus on a very important aspect, however it will be good if you also enumerate the predisposing factors which lead to the pregnancy complications and the outcomes.
--

REVIEWER	Gholizadeh, Leila University of Technology Sydney
REVIEW RETURNED	04-Oct-2022

GENERAL COMMENTS	Abstract Strengths and Limitations: add the word “independently” to statement 3. Page 4, line 19, Miscarriage, should start with a lower letter. Page 4, line 27, “even complications often considered less serious such as hyperemesis was associated with a 70% increased risk of developing rheumatoid arthritis Miscarriage” , need a full stop sign at the end. Some in text-citations have a full stop sign after parenthesis and some before. Page 4, line 32, the sentence “Despite there being a number of systematic reviews that have described the link between adverse complications during pregnancy and a multitude of outcomes. the awareness low as and there is need of robust synthesis”, should be rephrased. Page 4, line 58: The anticipated start is March 22. The date to be revised. Page 5, line 20: Researchers mentioned including systematic reviews and metal analysis as well as primary studies where applicable in their umbrella review. Given the definition of an umbrella review, how the researchers justify the inclusion of primary studies in their review. If the researchers aim to conduct a systematic review for some of exposures or outcomes mentioned in their protocol, how they aim to include the results in their umbrella review given that their conducted systematic review will not a be a peer reviewed research synthesis. Why the researchers do not consider excluding some of exposures or outcomes if there is no systematic review to synthesize the research on these variables? Page 5, line 48: Given that the proposed umbrella review aims to look at the associations between complicated pregnancy and future risk of a few health conditions, what is the researchers’ rationale for including interventional studies? Page 8, Line 25, Quality Assessment: the first sentence should be rephrased. Page 8, Line 25, Quality Assessment: Again given the purpose of the study which is to look at associations between variables, why researchers are considering including RCTs and Non-RCTs? Page 8, Line 25, Quality Assessment: Will the researchers exclude studies based on their quality? Page 8, Line 37, Update of existing reviews: Again, if the researchers aim to update some of the systematic reviews first, and then include the results in their umbrella review, it should be noted that some non-peer reviewed results will be included in the umbrella review. Why not the researchers consider excluding outdated systematic reviews. Page 8, Line 59. How will the second reviewers check the data extracted? Their role in data extraction needs to be clear. How many second reviewers will be involved? Researchers should avoid using short paragraphs throughout the
--

	protocol. Appendix Table 1 is located within the reference list.
--	---

VERSION 1 – AUTHOR RESPONSE

Reviewer: 1 comments and author's response

1: This umbrella review that would start in March 2022 aims to summarise systematic reviews evaluating the association between pregnancy complications and five long-term health conditions of the women. This topic is very important for public health. However, there are several issues that should be addressed very carefully.

We thank the reviewer for feedback towards improving the manuscript.

2: There are lots of grammatical errors throughout the manuscript (e.g. space, period, upper or lower case letters).

We thank the reviewer for the thorough comments. The document has been revised and the appropriate corrections have been made.

3: It is a bit difficult to follow as the orders of pregnancy complications and health conditions are inconsistent among the text and tables. In 'Objectives' (p5), 1) Metabolic health conditions, 2) Functional health conditions, 3) Autoimmune conditions, 4) Cancers, and 5) Mental health conditions are listed. However, Table 2 lists 'Mental health', 'Functional conditions',..., and Appendix Table 3 lists 'Autoimmune conditions', 'Metabolic health conditions'..., in order. Please keep consistency in order and terminology (e.g. 'Autoimmune conditions' (text) vs. 'Autoimmune disorders' (Table 2), 'Autoimmune health conditions' (Appendix Table 3)).

The order of the list of pregnancy complications and health conditions has been corrected and is consistent throughout the manuscript. We now use the term "autoimmune health conditions" in place of "autoimmune disorders" throughout the document.

4: 'Placental abruption' in Table 1 is not defined in Appendix Table 1.

Please note the number of appendix tables has been changed due to addition of an additional table of Prisma-P checklist and citation of the tables in ascending order

The definition of placental abruption has now been added to Appendix Table 1 (now appendix table 2). "Placental abruption is the early separation of a placenta from the lining of the uterus before completion of the second stage of labor. Placental abruption is also called abruptio placentae."

5: 'Irritable bowel syndrome' is categorised as 'Functional conditions' in Table 2 although it is listed as 'Autoimmune health conditions' in Appendix Table 3.

Apologies for any confusion this has created. In Table 2 (now appendix table 3) irritable bowel syndrome is categorised under functional conditions. Whereas, in Appendix Table 3 inflammatory bowel disease (which is the combined term for Crohn's disease and ulcerative colitis) is added as a search term in order to be as inclusive as possible. Inflammatory bowel disease has been added to Table 1 within the parentheses.

6: The order of 'Pre-eclampsia-early and late onset' to 'Gestational hypertension' should be rearranged according to that of 'Hypertensive disorders of pregnancy' in Table 1.

Thank you for pointing this out, the order has been corrected to maintain consistency.

7: [Appendix Table 1] 'PPH' should be spelled out in 'Obstetric/postpartum haemorrhage'.

We have now added "postpartum haemorrhage" (PPH) in full in Appendix Table 1.

I'm wondering why analysis is limited to 'singleton baby' in 'Mode of birth'. Is multiple birth more likely to be born with Caesarean section?

We did not mean to limit it to singleton birth and the definition has been upgraded to the multiple or singleton births.

Spaces are needed between each item if applicable. For example, 'Fetal growth restriction' should be positioned to the same level to its explanation.

Both foetal growth restriction and intra-uterine growth retardation have been moved to separate columns.

Is 'Or' required at the first in 'Puerperal psychosis'?

The word "Or" has been removed from the puerperal psychosis definition.

'Pelvic girdle pain (PGP)' could be exchanged in order for 'Obstetric cholestasis' according to Table "Pelvic girdle pain (PGP)" and "obstetric cholestasis" have been exchanged.

8:[Appendix Table 2] The items 'Placenta previa' and 'Placenta accreta and placenta percreta' do not correspond to their explanations.

Placenta previa, placenta accreta and placenta percreta are now presented in separate rows.

'Fetal growth retardation' listed in appendix table 1 could be added within the same square of 'Intra-uterine growth retardation/intra-uterine growth restriction.'

Fetal growth retardation added separately

17. Pelvic girdle pain' in Table 1 is not seen.

Pelvic girdle pain (PGP) has now been added.

Please note the number of appendix tables has been changed due to addition of an additional table of Prisma-P checklist and citation of the tables in ascending order.

9:Appendix Table 3] The order of disease listed should be rearranged exactly according to Table 2.

For example, 'Systemic sclerosis' should be positioned just below 'Sjogren's syndrome', (not 'Sjogern's').

The autoimmune health conditions have been rearranged according to Table 2.

"sjogren's syndrome" has been corrected.

'Addison's disease (in Autoimmune conditions)' and 'Oesophagus (in Cancers)' is not seen in Table 2.

In reverse, 'colorectal' in Table 2 is not described in this appendix.

'Addison's disease' and 'Oesophagus cancer' have been added to the respective tables.

In 'Mental health Conditions', 'non-affective psychosis' and 'neurotic disorders' in Table 2 are not defined, too.

In table 2, we have specified that non-affective psychosis includes schizophrenia, paranoia, and psychosis: not otherwise specified.

In table 2, we have specified that neurotic disorders include anxiety, post-traumatic stress disorder, and obsessive-compulsive disorders.

Reviewer: 2 comments and author's response

1:The manuscript, titled "The association of pregnancy complications/risk factors with the development of future health conditions in women: Overarching protocol for umbrella reviews," describes in detail how the authors will compile and present the current evidence and data on the topic highlighted in the title. The protocol is of the highest scientific quality, the methods are clearly described, and the topic is of paramount importance. The reviewer found no deficiencies or inaccuracies in the text. However, a discussion of possible limitations would be desirable. How do the authors deal with different populations, genetic backgrounds, women with specific dietary preferences, and combinations thereof? For example, the trace element selenium has been shown to be important in pregnancy outcomes, child development, and postpartum thyroiditis and depression. However, this effect may be limited to populations with low baseline intakes of selenium, such as in many parts of Asia, Europe, and Africa, but notably not in the U.S., where most high-quality studies are conducted. Often such specific interactions are not taken into account in regular reviews or meta-analyses, yet they may become apparent in an umbrella review. A few words about how the authors will handle such oddities would certainly strengthen both the protocol and the final study.

We thank the reviewer for their appreciation of the work.

While there may be differences in population characteristics it might not be possible to combine all results from the included studies. If we do combine them, we may present effect sizes according to certain population characteristics such as country/region to account for differences in nutritional status. We will try to consider all heterogeneity including clinical, methodological and statistical and will ensure this is covered in the discussion section of the manuscripts of the umbrella reviews.

Reviewer 3 comments and author's response

1:This study will focus on a very important aspect, however it will be good if you also enumerate the predisposing factors which lead to the pregnancy complications and the outcomes.

Thank you for your comment. In light of this comment, we have added the following to the background section :

"Apart from excessive weight gain, obesity, smoking, alcohol intake, and other lifestyle factors that are associated with pregnancy complications, other predisposing factors like polycystic ovarian syndrome (PCOS) and periodontal disease have also recently been identified as risk factors for pregnancy complications such as preterm birth, gestational diabetes, pre-eclampsia, and low birth weight."

Reviewer 4 comments and author's response

1:Abstract- Strengths and Limitations: add the word "independently" to statement 3.

Two reviewers will be involved in the data extraction. After the first reviewer has completed the data

extraction second reviewer will check the data extraction sheet and provide comments. Quality assessment of the reviews will be independently done by both reviewers. The statement has been changed to "Screening and the quality assessment will be independently performed by two reviewers".

2: line 19, Miscarriage, should start with a lower letter.

'Miscarriage' has now been changed to 'miscarriage'.

3:Page 4, line 27, "even complications often considered less serious such as hyperemesis was associated with a 70% increased risk of developing rheumatoid arthritis Miscarriage" , need a full stop sign at the end.

We have added a full stop to the end of this sentence:

"even complications that are often considered to be less serious such as hyperemesis was associated with a 70% increased risk of developing rheumatoid arthritis."

4:Some in text-citations have a full stop sign after parenthesis and some before.

The full stop is now placed after the parenthesis throughout the manuscript.

5:page 4, line 32, the sentence "Despite there being a number of systematic reviews that have described the link between adverse complications during pregnancy and a multitude of outcomes. the awareness low as and there is need of robust synthesis", should be rephrased.

We have reworded this section to:

"Some systematic reviews have shown associations of pregnancy complications with risk factors and development of health conditions, but there is a need for further research to synthesise this evidence, which can be used for improving maternal health post pregnancy."

6:Page 4, line 58: The anticipated start is March 22. The date to be revised.

In line with the Prospero registration, the start date is 1st April,2022.

7:Page 5, line 20: Researchers mentioned including systematic reviews and meta analysis as well as primary studies where applicable in their umbrella review. Given the definition of an umbrella review, how the researchers justify the inclusion of primary studies in their review. If the researchers aim to conduct a systematic review for some of exposures or outcomes mentioned in their protocol, how they aim to include the results in their umbrella review given that their conducted systematic review will not a be a peer reviewed research synthesis. Why the researchers do not consider excluding some of exposures or outcomes if there is no systematic review to synthesize the research on these variables?

Yes, we agree that according to the definition of the umbrella review it must only include systematic reviews. In this case, the primary studies will be included as part of the discussion or will be considered while updating the reviews. This part has been removed from the manuscript and further clarified.

8:Page 5, line 48: Given that the proposed umbrella review aims to look at the associations between complicated pregnancy and future risk of a few health conditions, what is the researchers' rationale for including interventional studies? As the purpose of an umbrella review is to identify and synthesise evidence from systematic reviews, we will include all relevant systematic reviews. Some of these systematic reviews may include interventional studies.

Any secondary analysis of looking at collected data in the interventional studies around exposure and outcome of interest will be included though it does not relate to the exposure being the intervention in the particular study.This has been explained in the manuscript

9:Page 8, Line 25, Quality Assessment: the first sentence should be rephrased.

The sentence has been rephrased to:

"The assessment of the quality of the reviews will be performed independently by two reviewers using the AMSTAR 2 tool. Any differences will be resolved by discussion and a third reviewer will be consulted if necessary."

10:Page 8, Line 25, Quality Assessment: Again given the purpose of the study which is to look at associations between variables, why researchers are considering including RCTs and Non-RCTs?

We will not include primary studies in our umbrella reviews, but some of the systematic reviews may include interventional studies. We have removed the phrase from the quality assessment section of the manuscript.

11:Page 8, Line 25, Quality Assessment: Will the researchers exclude studies based on their quality? The reviews will not be excluded even if they are rated as being low quality. Reviews that do not mention the quality assessment of primary studies might be excluded but this will be at the reviewers' discretion.This has been clarified in the manuscript.

12:Page 8, Line 37, Update of existing reviews: Again, if the researchers aim to update some of the systematic reviews first, and then include the results in their umbrella review, it should be noted that some non-peer reviewed results will be included in the umbrella review. Why not the researchers

consider excluding outdated systematic reviews.

There have been prior umbrella reviews that have updated the existing systematic reviews and presented the results. These were non peer reviewed results. Here the author will be responsible for the review. For example

Okoth K, Chandan JS, Marshall T, Thangaratinam S, Thomas GN, Nirantharakumar K, Adderley NJ. Association between the reproductive health of young women and cardiovascular disease in later life: umbrella review. *Bmj*. 2020 Oct 7;371.

On the other hand, few umbrella reviews have not updated the reviews and presented the results. For example

Hailes HP, Yu R, Danese A, Fazel S. Long-term outcomes of childhood sexual abuse: an umbrella review. *The Lancet Psychiatry*. 2019 Oct 1;6(10):830-9.

Authors will consider whether to update the systematic review depending on the recommended criteria and quality of reviews, the number of primary studies included etc.

13:Page 8, Line 59. How will the second reviewers check the data extracted? Their role in data extraction needs to be clear. How many second reviewers will be involved?.

Two reviewers will be involved in data extraction. After the first reviewer has completed the data extraction, the second reviewer will check the data extraction sheet and provide any comments. Any differences will be resolved by discussion and a third reviewer will be consulted if necessary.

The assessment of the quality of the reviews will be performed independently by both the reviewers using the AMSTAR 2 tool. Any differences will be resolved by discussion and a third reviewer will be consulted if necessary. This has been added to the manuscript

14:Researchers should avoid using short paragraphs throughout the protocol. Thank you for pointing this out.

The paragraphs have been modified accordingly

15:Appendix Table 1 is located within the reference list.

The table has been moved to the Appendix.

Please note the number of appendix tables has been changed due to addition of an additional table of Prisma-P checklist and citation of the tables in ascending order.

VERSION 2 – REVIEW

REVIEWER	Schomburg, Lutz Charité Universitätsmedizin Berlin, Institute for Exper. Endocrinology
REVIEW RETURNED	17-Nov-2022

GENERAL COMMENTS	My critique has been addressed. I do like the manuscript and the analysis plan, and I am looking forward to the umbrella review and the results that it will present. One small typo in the following sentence: "Both common (ie depression and anxiety) and serious (affective and non-affective psychotic disorders) associate with poor health in pregnancy and miscarriage, and low birth weight and ... (?)"
--

REVIEWER	Gholizadeh, Leila University of Technology Sydney
REVIEW RETURNED	22-Nov-2022

GENERAL COMMENTS	Thank you for comprehensively addressing the comments made in the first review. Please read the manuscript thoroughly for typo and punctuation errors.
--

VERSION 2 – AUTHOR RESPONSE

Reviewer 2 Comment

2. My critique has been addressed. I do like the manuscript and the analysis plan, and I am looking

forward to the umbrella review and the results that it will present. One small typo in the following sentence: "Both common (ie depression and anxiety) and serious (affective and non-affective psychotic disorders) associate with poor health in pregnancy and miscarriage, and low birth weight and ... (?)"

We thank the reviewer for the comment towards improving the manuscript.

The typo in the sentence has been corrected to "Both common mental health disorders (i.e. depression and anxiety) and serious mental health disorders (affective and non-affective psychotic disorders) associate with poor health in pregnancy."

Reviewer 4 comment

3. Thank you for comprehensively addressing the comments made in the first review. Please read the manuscript thoroughly for typo and punctuation errors. We thank the reviewer for the comment towards improving the manuscript.

The manuscript has been screened thoroughly for typos and punctuation errors and required corrections have been made.